# Kalman Filter-Based Differential Privacy Federated Learning Method

**Xiaohui Yang**  and **Zijian Dong** *

School of Cyberspace Security and Computer, Hebei University, Baoding 071000, China; yxh@hbu.edu.cn
* Correspondence: dongzijian1994@foxmail.com

**Abstract:** The data privacy leakage problem of federated learning has attracted widespread attention. Using differential privacy can protect the data privacy of each node in the federated learning, but adding noise to the model parameters will reduce the accuracy and convergence efficiency of the model. A Kalman Filter-based Differential Privacy Federated Learning Method (KDP-FL) has been proposed to solve this problem, which reduces the impact of the noise added on the model by Kalman filtering. Furthermore, the effectiveness of the proposed method is verified in the case of both Non-IID and IID data distributions. The experiments show that the accuracy of the proposed method is improved by 0.3–4.5% compared to differential privacy federated learning.

**Keywords:** federated learning; privacy protection; differential privacy; Kalman filter

## 1. Introduction

With the development of industry and the improvement in computing power, the application of artificial intelligence in various fields of people's lives has been becoming more popular, affecting many aspects, including medical care, logistics, face and voice recognition, etc. There is a wide application of machine learning and profound learning benefits from computing power and massive data. The traditional machine learning and deep learning model rely on data centers and require large amounts of high-quality data. With the increasing requirements of machine learning and deep learning in our lives, the requirements for data quantity and quality are also higher. In addition, in production and life, other companies and organizations need to pay to collect massive amounts of data. Aside from the cost, protecting users' privacy is also an important factor limiting the data obtained. To address these issues, Google proposed federated learning [1–3], which has attracted attention from scholars and organizations. Federated learning can train models using decentralized data, which significantly increases the amount and quality of data. On the other hand, the participating nodes are not required to provide their own local data in federated learning. They only need to deliver the training results to protect the user's data privacy. However, federated learning does not make the user's privacy completely invulnerable, and the user's privacy may be obtained through the shared model parameters [4,5].

Two schemes have been proposed to improve privacy protection in federated learning, one based on homomorphic encryption [6–9] and the other based on differential privacy [10–13]. However, these methods have difficulties in practical application due to their limitations. Differential privacy relies on adding noise to the original data or training results to protect privacy, and it may reduce the model's accuracy. Too much noise can make the model less accurate than expected. At the same time, too little noise will not achieve the protection of user privacy, making it challenging to balance privacy protection and accuracy [8]. Homomorphic encryption techniques rely on complex encryption and decryption computations, which may cost a large amount of time. At the same time, the different computing power of devices and the different time used in complex calculations situations can make the aggregation unsynchronized [14].

To solve the above problems of federated learning, Kalman filter-based differential privacy federated learning (KDP-FL) is proposed. The proposed scheme consists of two parts, using differential privacy to protect the gradient privacy in federated learning and using the Kalman filter to reduce the noise added to preserve privacy in order to improve the accuracy of the global model. The main contributions of this paper are as follows.

(1)  A multiparty cooperative model training scheme that protects data privacy is proposed.
(2)  A differential privacy federated learning algorithm is proposed to protect user data privacy by a differential privacy mechanism.
(3)  The noise introduced by the differential privacy mechanism is reduced by Kalman filtering to improve the model's accuracy.

## 2. Related Work

Federated learning has been gradually applied to protect the data privacy of all parties in scenarios where multiple parties cooperate in training and need to exchange data, such as traffic monitoring [15], edge computing [14], and IoT [16], to ensure that the original data is not leaked. However, privacy analysis attacks mean that federated learning also faces the risk of privacy leakage. It is essential to solve the problems of federated learning, especially gradient privacy protection, where data privacy is a particular concern. There are currently two privacy-preserving methods in federated learning: (1) differential privacy and (2) homomorphic encryption.

Abadi et al. [10] proposed a differential privacy deep learning method that combined machine learning or deep learning methods with differential privacy and proposed a new computational theory to obtain lower privacy consumption with smaller constraints, providing a basis for introducing differential privacy in federated learning. Hu et al. [11] used a block coordinate descent to optimize the objective function and differential privacy, which could protect gradient privacy but performed poorly in non-convex models.

Zhao et al. [12] used differential privacy federated learning for vehicle nets to train intelligent traffic management systems to protect vehicle location privacy but did not evaluate the accuracy under the influence of noise. Homomorphic encryption is a cryptographic method based on the computational complexity theory of mathematical puzzles, where the model parameters computed by each training node are aggregated and sent to the specified node after homomorphic encryption, and the model parameters obtained after decryption are the same as those obtained by direct aggregation without encryption, using this property to protect gradient privacy. Li et al. [13] used the chain aggregation method to protect gradient privacy, which improved the accuracy, but the chain aggregation approach introduced a large communication overhead. Fang et al. [8] used the Pilliar algorithm combined with federated learning to protect gradient privacy, but the Pilliar algorithm requires complex power operations, which introduced a large time overhead. Ma et al. [9] used the multi-key homomorphic encryption (MK-HE) to solve the collusion attack leakage problem brought by a single key, which improved privacy protection for federated learning. Zhang et al. [7] improved the encoding method during gradient transmission, reducing the computational overhead due to encryption but decreasing the accuracy.

To balance accuracy and differential privacy, this paper uses a differential privacy approach to protect gradient privacy while using Kalman filtering to reduce the impact of the noise on the accuracy.

## 3. Proposed Method

This section introduces the Kalman filtering-based differential privacy federated learning method, which contains a Differential Privacy mini-Batch Gradient Descent Algorithm and the Kalman reduction parameters aggregation method.

### 3.1. Method Overview

There are two types of nodes in federated learning, the Request Node (RN) and the Training Node (TN). The nodes that need others to assist in training the model are called the RN, which deliver task requests to other nodes and initialize or update the global model. The nodes that assist the RN in training the model are called the TN, which train the model using local data and provide the training results, which is also called the gradient, to the RN. There are $C$ TNs participating in training each round. The workflow of KDP-FL is shown in Figure 1. Federated learning contains $T$ rounds of global training. Global training includes task definition, model update, and local training. The RN defines the task and initializes the global model. Then, it loops the following steps to finish the global training:

(1) The RN updates the global model parameters and sends them to the TN.
(2) The TN trains the model locally and obtains the local training result for the $t$th round of training after $e$ iterations. This step is also called local training.
(3) The TN adds Gaussian noise to the training result and sends them to the RN.
(4) The RN uses Kalman filtering on all the training result to reduce the noise.
(5) The RN aggregates all the training results to obtain the global model parameters for the next round.

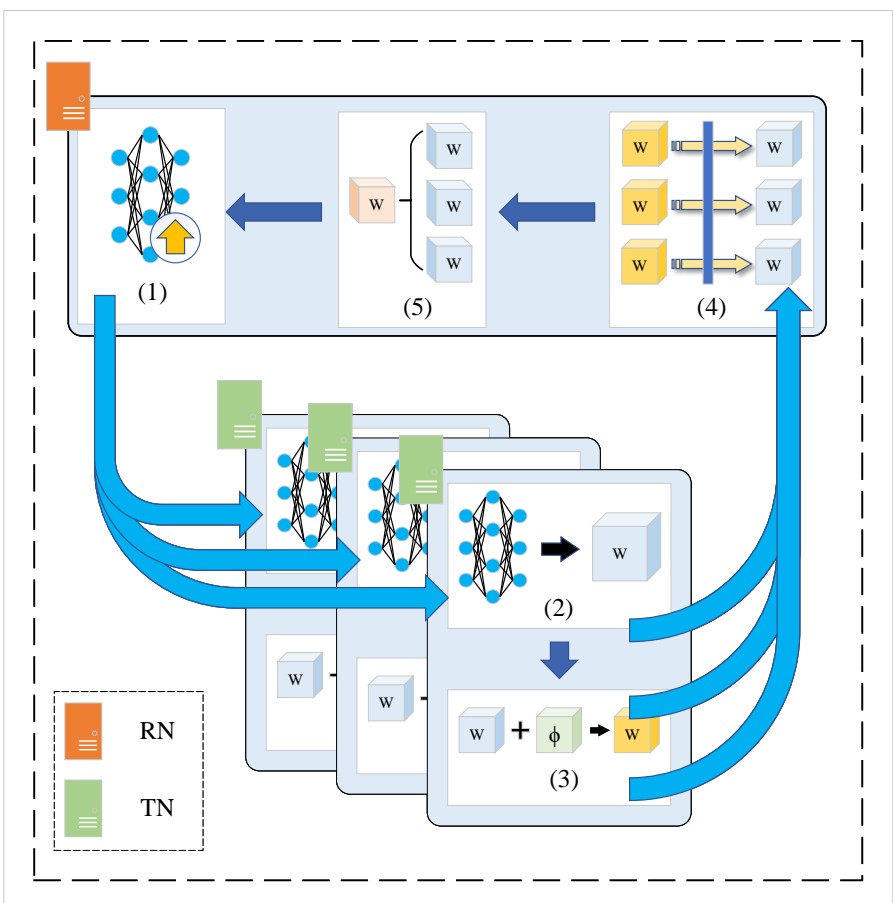

**Figure 1.** The workflow of KDPFL.

### 3.2. Method Details

In this section, we introduce our method in detail. Some significant symbols used in this paper are defined in Table 1.

**Table 1.** Symbol definitions.

| Symbol | Definition |
|---|---|
| $w_0$ | Initial global model parameters |
| $T$ | Total number of global training rounds |
| $E$ | The number of iterations performed in local training between two global training rounds |
| $\lambda$ | Learning rate |
| $B$ | Batch size |
| $g$ | Local training result of TN |
| $g'$ | Local training result of TN after adding noise |
| $w$ | Model parameters of TN in local training |
| $w_t$ | Global model parameters in the $t$th round of global training. |
| $w_{t,c}$ | Model parameters of TN$_c$ in the $t$th round of global training |
| $g_{t,c}$ | Local training result of TN$_c$ in the $t$th round of global training |
| $g'_{t,c}$ | Local training result of TN$_c$ in the $t$th round of global training after adding noise |
| $\mathcal{M}$ | Differential privacy mechanism |
| $D$ | Dataset |
| $M$ | Clipping threshold |
| $q$ | Sample probability |
| $I$ | Unit matrix |
| $C$ | The number of TN |

### 3.2.1. Differential Privacy-Based Stochastic Mini-BATCH Gradient Descent (DP-SGD)

TNs train the model locally instead of providing local raw data to the RNs to protect data privacy, then send the training result to the RN. Algorithm 1 describes the local training. Each TN uses $w_0$ as the initial local model parameters, uses the local data as the dataset, and trains the model using the DP-SGD algorithm. In the $e$th round of the DP-SGD, the TN performs several iterations to optimize the model parameters. During optimization, the training results are not sent to the RN, and noise is added to the training result at the end of the $E$th round of optimization of local training. Local training is performed using the following iteration

$$w = w - \lambda \frac{1}{B} \sum_{b=1}^{B} g_b \tag{1}$$

After $E$ rounds of local training, the training result should be sent to the RN. To protect the gradient privacy, the results need to be revised using a differential privacy method before sending. The differential privacy mechanism in this paper is the Gaussian mechanism, defined in Equation (2), where $\mathcal{N}\left(0, S_f^2 \cdot \sigma^2\right)$ denotes a normal distribution with mean zero and variance $S_f^2 \cdot \sigma^2$.

$$\mathcal{M}(D) = f(D) + \mathcal{N}\left(0, S_f^2 \cdot \sigma^2\right) \tag{2}$$

A Gaussian mechanism for a function $f$ with $S_f$ as sensitivity satisfies $(\varepsilon, \delta)$-differential privacy if $\delta \geq \frac{4}{5} \exp\left(-(\sigma\varepsilon)^2/2\right)$ and $\varepsilon < 1$ are satisfied [14]. Then, the local training result noise addition of the TN is calculated as:

$$g' = \frac{1}{\lambda}\left[w - w_0 + \mathcal{N}\left(0, \sigma^2 M^2 I\right)\right] \tag{3}$$

To address the gradient explosion that may occur during the training process, $L2$ norm gradient clipping is used. In addition, gradient clipping can limit the gradient values to a certain range, which is beneficial to the Gaussian mechanism. The clipping threshold is set to $M$. The gradient value $g$ does not change when the $L2$ norm number of the gradient $\|g\|_2 \leq M$, and if $\|g\|_2 > M$, the gradient is clipped as Algorithm 1 described. The gradient clipping method can be used to control the gradient values from exceeding the set threshold and prevent the impact of gradient explosion on the model. In addition, after gradient

clipping, the upper bound of the gradient value $L2$ norm is $\|g\|_2 \leq M$. Limiting the upper bound can effectively control the variance in the noise distribution.

The use of differential privacy in SGD requires the calculation of privacy consumption, and there is a risk of privacy leakage when the privacy cost is higher than the privacy budget. Compared to the strong composition theorem [17], using the Moment Accountant proposed by Adabi et al. [10] can obtain a smaller $\sigma$ with the same privacy budget. According to the theory of Moment Account, for any $\varepsilon < c_1 q^2 T$ and $\sigma \geq c_2 \frac{q\sqrt{T \log(1/\delta)}}{\varepsilon}$, the DP-BGD algorithm is $(\varepsilon, \delta)$-differentially private for any $\delta > 0$ where $c_1, c_2$ are constants, and $q$ is the sample probability defined in Equation (4). The variance in the added noise is based on the number of training rounds and privacy budget.

$$q = \frac{C}{N} \frac{BE}{|D|} \tag{4}$$

$$\sigma \geq c_2 \frac{q\sqrt{T \log(1/\delta)}}{\varepsilon} \tag{5}$$

The DP-SGD algorithm is shown in Algorithm 1.

---

**Algorithm 1** DP-SGD

---

**Input:** $D_c$; $w_0$, $M$;
**Output:** $g'$
1:  $w \leftarrow w_0$
2: **for** $e = 1$ **to** $E$ **do**
3:     Take a random batch with size $B$
4:     **for** $b = 1$ **to** $B$ **do**
5:        $g_b \leftarrow \frac{\partial l(w, x_b)}{\partial w}$
6:        $g_b \leftarrow g_b / \max\left(1, \frac{\|g_b\|}{M}\right)$
7:     **end for**
8:     $w \leftarrow w - \lambda \frac{1}{B} \sum_{b=1}^{B} g_b$
9: **end for**
10: $g' \leftarrow \frac{1}{\lambda}\left[w - w_0 + \mathcal{N}\left(0, \sigma^2 M^2 I\right)\right]$

---

### 3.2.2. Kalman Noise Reduction Parameter Aggregation

The TN sends the training result to the RN after adding noise using Equation (3), and the model parameters of node $TN_c$ are noted as $g_c$; then, the RN receives the set of model parameters $G = \{g'_1, g'_2, \cdots, g'_c\}$.

The model parameters obtained by the RN contain noise, which will decrease the accuracy of the model. In order to reduce the impact of differential privacy on the model and improve the accuracy, it is necessary to revise the model parameters sent by the TN. In this paper, we use Kalman filtering to revise all model parameters obtained by the RN to improve the accuracy of the model under the situation where differential privacy noise has been added to the global model:

$$g = Kalman(g') \quad g' \in G \tag{6}$$

The details will be described in Section 3.2.3. The global model parameters of the *t*th round are obtained after aggregating all the model parameters revised by the Kalman filter; then, the global model is updated.

$$w_{t+1} = w_t - \frac{\lambda}{C} \sum_{c=1}^{C} g_{t,c} \tag{7}$$

The KDP-FL algorithm is shown in Algorithm 2.

---

**Algorithm 2** KDP-FedAvg

---

**Input:** $D_i; w_0;$
**Output:** $w$
 1: **for** $t = 1$ **to** $T$ **do**
 2:    **for** $c = 1$ **to** $C$ **in parallel do**
 3:       TN$_c$ **do:** $g'_{t,c} \leftarrow$ DP-SGD$(D_i, w, M)$
 4:       RN **do:** $g_{t,c} \leftarrow$ Kalman$\left(g'_{t,c}\right)$
 5:    **end for**
 6:    $w_{t+1} \leftarrow w_t + \frac{\lambda}{C} \sum_{c \in C} g_{t,c}$
 7: **end for**

---

### 3.2.3. Details of Kalman Noise Reduction

The Kalman filter is able to reduce the noise of normal distribution in the linear system [18]. The noise added to the training result after differential privacy is Gaussian noise satisfying the condition of normal distribution, and the state update equation is linear, so the Kalman filter can be used for noise reduction in the training result to decrease the impact of differential privacy on the accuracy of the model. The noise revising has two steps, the first step is the model parameters' update, and the second step is the parameters' update correction.

There are $C$ TNs in the $t$th round of federated learning; the model parameters of the TN$_c$ are $w_{t,c}$, the global model parameters are $w_t$, the parameter vector composed of the model parameters of all TNs is $W_t = \begin{bmatrix} w_t & w_{t,1} & w_{t,2} & \cdots & w_{t,N} \end{bmatrix}$, and the gradient values of TN$_c$ are $g_{t,c}$. Then, the gradient vector composed of the gradient values of all TNs is $G'_t = \begin{bmatrix} 0 & g'_{t,1} & g'_{t,2} & \cdots & g'_{t,c} \end{bmatrix}$. The linear coefficient matrix of the state prediction equation $U$ is a unit matrix, and $V$ is defined as following:

$$V = \begin{bmatrix} 1 & 0 & 0 & \ldots & 0 \\ 0 & \frac{\lambda}{C} & 0 & \ldots & 0 \\ 0 & 0 & \frac{\lambda}{C} & \ldots & 0 \\ \ldots & \ldots & \ldots & \ddots & \ldots \\ 0 & 0 & 0 & \ldots & \frac{\lambda}{C} \end{bmatrix} \tag{8}$$

Then, the parameter update equation for model training can be transformed into the state prediction equation:

$$W_{t+1}^T = UW_t^T + VG'_t^T \tag{9}$$

In fact, the parameter update equation Equation (9) for model training can also be written as:

$$\begin{cases} w_{t+1} = w_t - \lambda \sum_{c \in C} g'_{t,c} \\ w_{t+1,1} = w_{t,1} - \lambda g_{t,1} + \mathcal{N}\left(0, \sigma^2 M^2 I\right) \\ \ldots \\ w_{t+1,c} = wt, c - \lambda g_{t,c} + \mathcal{N}\left(0, \sigma^2 M^2 I\right) \end{cases}, \tag{10}$$

The noise added by the TN in differential privacy uses the Gaussian mechanism satisfying Gaussian distribution.

To achieve dimensional matching with $U$ and $V$, we need the calculation described in Algorithm 3. The elements in $W_t^T$ and $G'_t^T$ are all matrix. First, we reshape each element in $W_t^T$ and $G'_t^T$. Let $N = m \times n$, and each matrix $w_{t,c}$ or $g'_{t,c}$ shaped $m \times n$ will be reshaped to $N \times 1$. Let $\omega_{c,i}$ be the $i$th parameter in $w_{t,c}$ and $\varrho_{c,i}$ be the $i$th in $g'_{t,c}$. Then, we can calculate Equation (9) using Algorithm 3.

---

**Algorithm 3** Matrix broadcast calculation

---

**Input:** $G_t'^T; W_t^T; U; V;$
**Output:** $W_{t+1}$
 1: **for** $i = 1$ **to** $N$ **do**
 2:     $\omega_i \leftarrow \begin{bmatrix} \varpi_i & \varpi_{1,i} & \varpi_{2,i} & \cdots & \varpi_{C,i} \end{bmatrix}$
 3:     $\rho_i \leftarrow \begin{bmatrix} 0 & \varrho_{1,i} & \varrho_{2,i} & \cdots & \varrho_{C,i} \end{bmatrix}$
 4:     $\omega_i^T \leftarrow U\omega_i^T + V\rho_i^T$
 5: **end for**
 6: $w_{t+1} \leftarrow \begin{bmatrix} \varpi_1 & \varpi_2 & \cdots & \varpi_N \end{bmatrix}$
 7: **for** $c = 1$ **to** $C$ **do**
 8:     $w_{t+1,c} \leftarrow \begin{bmatrix} \varpi_{c,1} & \varpi_{c,2} & \cdots & \varpi_{c,N} \end{bmatrix}$
 9: **end for**
10: $W_{t+1} \leftarrow \begin{bmatrix} w_{t+1} & w_{t+1,1} & w_{t+1,2} & \cdots & w_{t+1,C} \end{bmatrix}$

---

The gradient vector $G_k'^T$ contains noise added by the TN, and the distribution of the noise is known. The Gaussian noise added by the differential privacy is considered an external disturbance, i.e., uncertainty in the system.

In Kalman filtering, $z_t$ is used to denote the actually obtained data, calculated using the following equation:

$$z_t = HW_t + R, \tag{11}$$

where $H$ is the data unit transformation matrix. In this paper, the data sent by the TN are the same as the data obtained by the RN; so, $H$ is the $C + 1$ dimensional unit vector. $R$ is the error column vector. In order to avoid the occurrence of a singular matrix in subsequent calculations, here, $R = 0.1Q$ is taken. The calculation of the Kalman gain matrix using the noise covariance matrix $P_t$ is

$$K_{t+1} = HP_tH^T\left(HP_tH^T + R\right)^{-1} \tag{12}$$

Since $H$ is a $C + 1$ dimensional unit vector, the formula for the Kalman gain can be reduced to

$$K_{t+1} = P_t(P_t + R)^{-1}. \tag{13}$$

The parameter update equation using the Kalman gain correction is

$$W_{t+1} = W_t + K_t(z_t - HW_t). \tag{14}$$

We update the covariance matrix of the norm noise:

$$P_{t+1} = (I - K_tH)P_t + Q. \tag{15}$$

$$Q \sim \mathcal{N}\left(0, \sigma^2 M^2 I\right) \tag{16}$$

## 4. Experiment and Analysis

### 4.1. Experiment Setting

The public datasets MNIST [19], FMNIST [20], and CIFAR10 [21] were selected to train and test. The dataset was partitioned to simulate a federated learning environment, (1) independently and identically distributed (IID) for each TN [1]: the training data were shuffled and then partitioned into 100 clients each receiving an equal amount of examples; (2) non-independently and identically distributed (Non-IID) for each TN [22]: we distributed the data among 100 TNs, such that each TN contained samples of only two kinds; the number of samples among the TN followed a power law, as detailed in Table 2.

**Table 2.** Dataset details.

| Dataset | Labels | Devices | Training Samples | Samples/Device | |
| --- | --- | --- | --- | --- | --- |
| | | | | mean | std |
| MNIST | 10 | 100 | 60,000 | 600 | 328 |
| FMNIST | 10 | 100 | 60,000 | 600 | 328 |
| Cifar10 | 10 | 100 | 50,000 | 500 | 293 |

Using CNN as the training model for MNIST and FMNIST, the network architecture consisted of two fully connected layers. The convolutional layers used $5 \times 5$ convolutions with stride 1, followed by a ReLU and $2 \times 2$ max pools, with 32 channels in the first layer and 64 channels in the second layer. Then the first convolution output was $32 \times 14 \times 14$ after pools, and the second convolution output was $64 \times 7 \times 7$ after pools. Last, we had two fully connected layers and used resnet50 [23] to train the model for CIFAR10.

The experimental environment was an Intel(R) Xeon(R) W-2295 CPU@3.00 GHz, 64 GB RAM, NVIDIA GeForce RTX 3080 GPU, and the experimental platform was Pytorch. According to the related works [10–12], the parameters of this paper were set as shown in Table 3.

**Table 3.** Parameter definitions.

| Definition | Parameter |
| --- | --- |
| Total number of nodes | $C_{all} = 100$ |
| Number of TNs involved in training | $C = 20$ |
| Number of local batches | $B = 32$ |
| Learning rate | $\lambda = 0.05$ |
| Local training round | $E = 5$ |
| Privacy budget | $\varepsilon = 1$ or $\varepsilon = 5$ |

### 4.2. Privacy Budget Analysis

Moments Accountant can provide a tighter bound than the composition theorem on privacy loss. The privacy loss can be calculated by the noise level $\sigma$. The privacy loss with the increasing global epoch is shown in Figure 2. To test the effectiveness of the proposed method under different privacy budgets, we selected $\varepsilon = 1$ and $\varepsilon = 5$. A larger privacy budget $\varepsilon = 5$ will result in a more accurate model, and a smaller privacy budget $\varepsilon = 1$ will achieve better privacy protection. Then, we let $\delta = 10^{-5}$ and computed the noise level $\sigma$ using Equation (5).

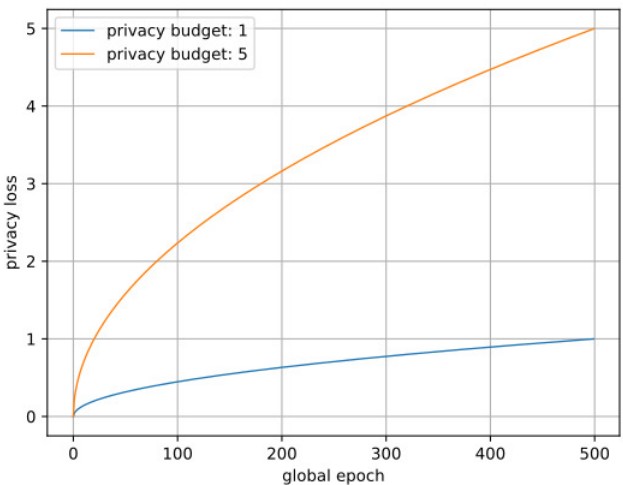

**Figure 2.** Privacy loss with increasing global epoch.

### 4.3. Experimental Results

The effectiveness of the method is shown by the model accuracy, and the comparison results are shown in Figure 3. The DP-FedAvg algorithm protected privacy by adding noise to the gradient matrix, and the impact of privacy protection depended on the size of the privacy budget. The smaller the privacy budget, the better the privacy protection. However, as the privacy budget decreased, the noise required to be added increased, which may lead to a decrease in the accuracy of the model. By comparing the accuracy under the non-IID dataset, DP-FedAvg in (a) and (c), a higher requirement for privacy protection needed a smaller $\varepsilon$, which led to lower accuracy of the model. Furthermore, the accuracy of DP-FedAvg at $\varepsilon = 1$ decreased about 3% compared to that at $\varepsilon = 5$. Furthermore, the accuracy of DP-FedAvg $\varepsilon = 1$ was lower than $\varepsilon = 5$ and was the same as the pattern in the IID.

In addition, using different datasets, FMNIST and CIFAR10, the experimental results in Table 4 show that KDPFL achieved good results in the other two datasets. Hence, the model is suitable for other datasets. By testing CNN and CIFAR10, the experimental results show that KDPFL is suitable for different neural network structures and has good compatibility.

**Table 4.** Accuracy in different models and datasets.

| Method | FMNIST (acc/%) | | | | CIFAR10 (acc/%) | | | |
|---|---|---|---|---|---|---|---|---|
| | Non IID | | IID | | Non IID | | IID | |
| | $\varepsilon = 1$ | $\varepsilon = 5$ | $\varepsilon = 1$ | $\varepsilon = 5$ | $\varepsilon = 1$ | $\varepsilon = 5$ | $\varepsilon = 1$ | $\varepsilon = 5$ |
| FedAvg | 86.62 | | 89.56 | | 83.21 | | 86.34 | |
| DP-FedAvg | 83.17 | 84.94 | 86.59 | 88.02 | 77.26 | 79.44 | 81.23 | 83.55 |
| KDPFL | 86.12 | 86.23 | 88.93 | 89.01 | 82.76 | 82.88 | 85.93 | 86.03 |

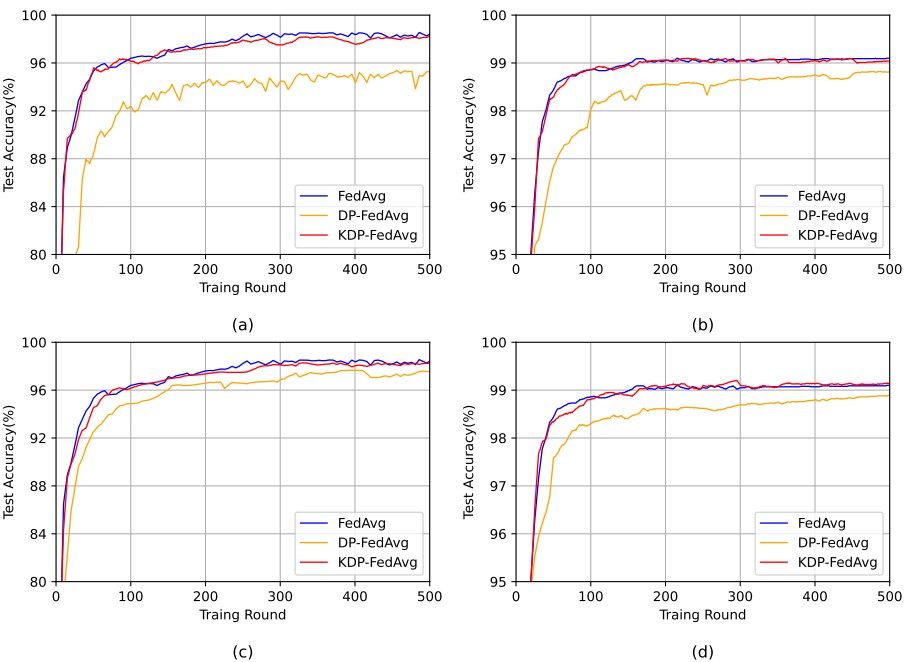

**Figure 3.** Experiment results: (**a**) non-IID, $\varepsilon = 1$; (**b**) IID, $\varepsilon = 1$; (**c**) non-IID, $\varepsilon = 5$; (**d**) IID, $\varepsilon = 5$.

In this paper, the parameter update equation was transformed into the state update equation. Furthermore, the global model parameter update, which was completely determined by the TN's training result, was transformed into the global model parameter update, which was jointly determined by the estimated value and the TN calculation result. The Kalman gain coefficient was calculated by the noise covariance matrix, and the weight

of the estimated value and the predicted value in the global model update were calculated by the Kalman gain coefficient to realize the optimal global model update.

The Kalman filter noise reduction method, which reduced the noise at aggregation, decreased the effect of the differential privacy added noise on the model, achieved an accuracy closer to the FedAvg algorithm without added noise, and performed well with both non-IID data and IID data. With non-IID data, the accuracy $\varepsilon = 1$ was improved up to 4.5% compared to DP-FedAvg, and the accuracy $\varepsilon = 5$ was improved up to 2.5%. The above experiments show that the scheme proposed in this paper can protect the user's privacy with little loss of accuracy.

## 5. Conclusions

In this paper, a Kalman filter-based differential privacy federated learning method, KDP-FL, was proposed. Federated learning was used to achieve data sharing while preserving privacy. In addition, differential privacy was added to federated learning, which can resist gradient differential attacks and further protect user privacy. Furthermore, the influence of the noise added by the differential mechanism on the accuracy of the model was reduced by the Kalman filter. The experiment results and analysis showed that the proposed method was able to train models with little loss of accuracy while protecting user privacy. KDP-FL is suitable for training privacy-preserving models under multiple nodes and can help in data cooperation among multiple nodes.

When the neural network is simple, the additional computational overhead brought by Algorithm 3 is negligible. However, when the neural network is complex with a large number of parameters, optimizing all parameters requires calculating the Kalman gain for each parameter, which will bring about a non-negligible additional computational overhead.

In addition, the proposed method used Kalman filtering, which increases the computational overhead and is not applicable to devices and nodes with weak computational power, which is a direction for future improvement.

**Author Contributions:** Conceptualization, X.Y. and Z.D.; data collection and analysis, Z.D.; validation, X.Y.; writing—original draft preparation, Z.D.; writing—review and editing, X.Y. and Z.D. All authors have read and agreed to the published version of the manuscript.

**Funding:** This work was supported by Ministry of Science and Technology of China, National Key R&D Program "Cyberspace Security" Key Project, 2017YFB0802305 and the Natural Science Foundation of Hebei Province, F2021201052.

**Institutional Review Board Statement:** Not applicable.

**Informed Consent Statement:** Not applicable.

**Data Availability Statement:** Publicly available datasets were analyzed in this study. The source code can be found here: https://git.acwing.com/dzj/kdpfl.git (accessed on 29 July 2022), and the datasets can be found here: https://pan.baidu.com/s/1PVFJ7vnezGe9jEp8Pgh_fg?pwd=n2w5 (accessed on 29 July 2022).

**Conflicts of Interest:** The authors declare no conflicts of interest.

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
