# Peer review of "Kalman Filter-Based Differential Privacy Federated Learning Method"

_applsci, doi:10.3390/app12157787_

Round 1

Reviewer 1 Report

This paper describes an improved method to conduct federal learning with privacy constraints. Using the framework of differential privacy, where each node involved in federated learning adds noise to the model parameters before transmitting results to the central node (here called request node), the authors propose to use a Kalman filter after each iteration to diminish the loss caused by the noise without affecting the privacy guarantees.

After an introduction and a description of different methods to ensure privacy in federated learning, the authors describe their method in section 3 and conduct an experiment to test how the Kalman filter improves accuracy in Section 4.

In general, this article seems sound and the experiment conducted in Section (4) supports the premises and the conculsions of the article adequately, by indicating that the use of a Kalman filter indeed imporves accuracy. The idea to use a Kalman filter in these circumstances also seems to be justified, as the Kalman filter takes advantage of the repeated queries of federated learning without affecting privacy guarantees, because this kind of inference is implicitly taken into account in the computation of the privacy budget.

However, the general descriptions of the method and of the experiment setting are very confusing. The general ideas of the paper are clear, but understanding the details is harder, and as a result it is hard to assess the scientific soundness of the paper thoroughly  (in particular, I am not sure the relation between \sigma,  \epsilon and \delta is correct as it is written). Section 3.2 and the first paragraph of section 4.1 (l.134-140) should be edited to make these details clearer. These improvements should address (but not necessarily be limited to) the following remarks:

1) The distinction between local training and federated learing is not always clear, which generates confusion about indices

- the index for a generic node is sometimes i, sometimes c

- the index for the final node is sometimes c, sometimes C, sometimes N

- in some equations there is a node with index 0, sometimes the first index is 1

- the index of the iteration is sometimes i, sometimes k

- when the authors tackle the fact that model parameters on a given node (and at a given iteration k), a new index j is introduces, and algorithm 3 describes how each parameter is computed separately, but in this context, it is strange to take a fixed value of i in l.1, and then present l.2-4 in an array fashion without a mention of i, and l.6 of Algorithm 3 does not make sense, because j is not defined out of the loop, and logically, W_k should compile the vectors found for different values of j.

In general, the authors need to give clearer indication of the role of each variable or index when it is introduced.

- In section 3.2.1, Algorithm 1 describes a round of local training with gradient descent, |T| is implied to be the number of local iterations ;In algorithm 2, the reader understands that e \in E represents the round of global training, but E is never explicitly described, and the role of e is only mentioned quickly. However, T is also described as the number of global training rounds, and in the experiment E=5 is described as 'Local training round'

Logically, both E and T should appear in the relation fixing the value of \sigma for a given pair (\varepsilon,\delta). Here, only T appears and it seems starnge that E plays no role in this relation.

2) The description of the description of the IID and non IID datasets is insufficiently clear:

- IID 'Independently and identically distributed (IID) for each node, all training samples are randomly divided into 100 parts, and each node has equal amount of samples' : does that mean that the dataset is divided in 100 sub-datasets of equal size in an equiprobable fashion? If so, the qualificiation IID is not necessarily appropriate, as the affectations of the samples are dependent (to ensure equal size).

- Non-IID, no indication is given on the probability law with which samples are attributed to nodes, except for the constraint : 'each node having two kinds of labeled data'. Does that mean that each node must have exactly 2 different labels? at least 2 different labels? Does this condition mean something else?

The English language and style is generally acceptable ; however, several sentences have an unusual structure, which hinders the understanding (before eq (12), before eq (15), l.141, l.146 for example), sometimes punctuation is missing (no point between two sentences, or a superfluous point in the middle of a sentence).

I have a more general interrogation regarding the guarantees of differential privacy in federated learning. In general, the strong guarantees of differential privacy are ensured for a given dataset by giving answers to a finite number of requests, and refusing further requests when the privacy budget is reached. Here, even if differential privacy is reached for a given task involving federated learning, how can one be sure that no further request by another federated learning task will lead to privacy breaches. Is there a mean to preserve the utility of a dataset that has been used in a federated learning task in a differential privacy compliant manner?

Here are more detailed remarks:

l.17-18 : what is meant by the expression ‘in production and life’ ?

l.18 : the data quantity and quality are also higher requirements : the requirements on data quantity and quality are also higher

Figure (1) : the five steps of the cycle, and especially steps (3), (4) and (5) correspond more to individual diagrams than to the arrows between these diagrams. As a consequence, I would advise enclosing each of these individual diagrams in a small box and attributing numbers to these boxes rather than attributing numbers to arrows.

l.95 : RN public the task : I do not understand what public (as a verb) means. Does this sentence means ‘RN defines the task’, ‘RN makes the task public’, or something else ?

Un-numbered line after l. 106 : […] defined in Eq. 3, which […] → […] defined in Eq.3, where […] ; after the mathematical expression of the noise, the comma should be on the same line, not at the beginning of the new line.

In equation (2), I don’t understand what \mathcal{M}(d) represents ; is it a noisy version of f ?

Last line of p.5 : W_k=[w_k w_{k,1},…, w_{k,N}]., : delete the point before the comma ; there should be a w_{k,0} between w_k and w_{k,1} as there is a node 0 before ; these notation for W is not coherent with l.127, where there is no global parameter w, only parameters associated with nodes.

There seems to several errors in equations (8) and (9) ; the coefficient C should not apply to the whole matrix ; as it is we would have w_{k,i}=1/C * w_{k-1,i} + g_{k-1}, which is not coherent. The factor \lambda does not appear in equations (8) and (9) ; as a result, eq (9) as it is written is not coherent witn eq (1), (6) and (7). If the factor \lambda and the noise are included in G_{k,i}, it should be stated explicitly ; in any case, noting g_{k,i} the quantity ‘-\lambda g_i(k) + noise’ is confusing.

l.130 : Elements in … and … are all matrix : are the w_{k,i} matrices or vectors ?

l.130 : Let w_{k,i}(j) be the ith element… : ith should be jth (if I understand the intention properly)

line before eq (10) : there is a superfluous point after noise

l.150 should be Privacy budget analysis

Figure 2 and Figure 3 hide part of the text above them

Author Response

Dear reviewer:

Thank you very much for your comment and professional advise. The opinions help to improve academic rigor of our article. Based on your suggestion, we have made corrected modifications on the revised manuscript. We hope our work can be improved again. Furthermore, we list the advice and then response. Besides, we also upload a pdf version.

Opinions:

  1. However, the general descriptions of the method and of the experiment setting are very confusing. The general ideas of the paper are clear, but understanding the details is harder, and as a result it is hard to assess the scientific soundness of the paper thoroughly (in particular, I am not sure the relation between \sigma, \epsilon and \delta is correct as it is written). Section 3.2 and the first paragraph of section 4.1 (l.134-140) should be edited to make these details clearer. These improvements should address (but not necessarily be limited to) the following remarks:

1.1 The distinction between local training and federated learning is not always clear, which generates confusion about indices.

1.1.1 The index for a generic node is sometimes i, sometimes c

1.1.2 The index for the final node is sometimes c, sometimes C, sometimes N

1.1.3 In some equations there is a node with index 0, sometimes the first index is 1

1.1.4 The index of the iteration is sometimes i, sometimes k

1.1.5 When the authors tackle the fact that model parameters on a given node (and at a given iteration k), a new index j is introduces, and algorithm 3 describes how each parameter is computed separately, but in this context, it is strange to take a fixed value of i in l.1, and then present l.2-4 in an array fashion without a mention of i, and l.6 of Algorithm 3 does not make sense, because j is not defined out of the loop, and logically, W_k should compile the vectors found for different values of j. In general, the authors need to give clearer indication of the role of each variable or index when it is introduced.

1.1.6 In section 3.2.1, Algorithm 1 describes a round of local training with gradient descent, |T| is implied to be the number of local iterations ;In algorithm 2, the reader understands that e \in E represents the round of global training, but E is never explicitly described, and the role of e is only mentioned quickly. However, T is also described as the number of global training rounds, and in the experiment E=5 is described as 'Local training round'. Logically, both E and T should appear in the relation fixing the value of \sigma for a given pair (\varepsilon,\delta). Here, only T appears and it seems strange that E plays no role in this relation.

1.2 The description of the description of the IID and non IID datasets is insufficiently clear:

1.2.1 IID 'Independently and identically distributed (IID) for each node, all training samples are randomly divided into 100 parts, and each node has equal amount of samples' : does that mean that the dataset is divided in 100 sub-datasets of equal size in an equiprobable fashion? If so, the qualificiation IID is not necessarily appropriate, as the affectations of the samples are dependent (to ensure equal size).

1.2.2 Non-IID, no indication is given on the probability law with which samples are attributed to nodes, except for the constraint : 'each node having two kinds of labeled data'. Does that mean that each node must have exactly 2 different labels? at least 2 different labels? Does this condition mean something else?

  1.  

2.1 The English language and style is generally acceptable ; however, several sentences have an unusual structure, which hinders the understanding (before eq (12), before eq (15), l.141, l.146 for example), sometimes punctuation is missing (no point between two sentences, or a superfluous point in the middle of a sentence).

2.2 I have a more general interrogation regarding the guarantees of differential privacy in federated learning. In general, the strong guarantees of differential privacy are ensured for a given dataset by giving answers to a finite number of requests, and refusing further requests when the privacy budget is reached. Here, even if differential privacy is reached for a given task involving federated learning, how can one be sure that no further request by another federated learning task will lead to privacy breaches. Is there a mean to preserve the utility of a dataset that has been used in a federated learning task in a differential privacy compliant manner?

2.3 Here are more detailed remarks:

2.3.1 l.17-18 : what is meant by the expression ‘in production and life’ ?

2.3.2 l.18 : the data quantity and quality are also higher requirements : the requirements on data quantity and quality are also higher.

2.3.3 Figure (1) : the five steps of the cycle, and especially steps (3), (4) and (5) correspond more to individual diagrams than to the arrows between these diagrams. As a consequence, I would advise enclosing each of these individual diagrams in a small box and attributing numbers to these boxes rather than attributing numbers to arrows.

2.3.4 l.95 : RN public the task : I do not understand what public (as a verb) means. Does this sentence means ‘RN defines the task’, ‘RN makes the task public’, or something else ?

2.3.5 Un-numbered line after l. 106 : […] defined in Eq. 3, which […] → […] defined in Eq.3, where […] ; after the mathematical expression of the noise, the comma should be on the same line, not at the beginning of the new line.

2.3.6 In equation (2), I don’t understand what \mathcal{M}(d) represents ; is it a noisy version of f ?

2.3.7 Last line of p.5 : W_k=[w_k w_{k,1},…, w_{k,N}]., : delete the point before the comma ; there should be a w_{k,0} between w_k and w_{k,1} as there is a node 0 before ; these notation for W is not coherent with l.127, where there is no global parameter w, only parameters associated with nodes.

2.3.8 There seems to several errors in equations (8) and (9) ; the coefficient C should not apply to the whole matrix ; as it is we would have w_{k,i}=1/C * w_{k-1,i} + g_{k-1}, which is not coherent. The factor \lambda does not appear in equations (8) and (9) ; as a result, eq (9) as it is written is not coherent witn eq (1), (6) and (7). If the factor \lambda and the noise are included in G_{k,i}, it should be stated explicitly ; in any case, noting g_{k,i} the quantity ‘-\lambda g_i(k) + noise’ is confusing.

2.3.9 l.130 : Elements in … and … are all matrix : are the w_{k,i} matrices or vectors ?

2.3.10 l.130 : Let w_{k,i}(j) be the ith element… : ith should be jth (if I understand the intention properly)

2.3.11 line before eq (10) : there is a superfluous point after noise

2.3.12 l.150 should be Privacy budget analysis

2.3.13 Figure 2 and Figure 3 hide part of the text above them.

Authors response:

1

1.1 We add some describe in our new manuscript in l.96 and l.102 to make the distinction between local training and federated learning clear.

1.1.1 We modified the generic node index to c, we call node c TNc.

1.1.2 We modified all TN indexes to 1,2,...,c,...,C. The first index is TN1,and the last one is TNC.

1.1.3 We modified all euations such that index begin with 1.

1.1.4 We reduced some unnecessary indexes and fixed indexes with specific meanings, using b as the index of the data in the batch, t as the global training index, and e as the local training index. For the temporary index, we use i.

1.1.5 We removed some unnecessary indices in Algorithm 3 and used i as the index of the elements in the gradient matrix after reshape. Use c as the index to traverse all TNs and t as the index for the number of global training epochs. The added variable is defined in l.152 to l.156.

In addition, we have added a symbol description table at the beginning of page 4 to clarify the role of each variable or index.

1.1.6 We redefine T and E, where T is the number of global training epochs and E is the number of local training epochs, and added a description of the roles of T and E in Table 1 on page 4.

Sorry we left out the definition of q, in the new manuscript we have added the definition of q. q contains E, so both T and E work in this relation.

1.2

1.2.1 The processing method of IID refers to the literature “McMahan, B.; Moore, E.; Ramage, D.; Hampson, S.; y Arcas, B.A. Communication-Efficient Learning of Deep Networks from Decentralized Data. In Proceedings of the AISTATS. PMLR, 2017, Vol. 54, Proceedings of Machine Learning Research, pp.1273–1282.”, but as you said, the description is not clear enough, resulting in some ambiguity, and we modified the expression.

1.2.2 Our descriptions are not detailed enough to make readers clear,  and we have revised our description. There are many cases that are Non-IID. Here we refer to the practice in “Li, X.; Huang, K.; Yang, W.; Wang, S.; Zhang, Z. On the Convergence of FedAvg on Non-IID Data. In Proceedings of the ICLR. OpenReview.net, 2020.”. The choice of two classes is based on extreme cases. This situation has a serious impact on the model. In this case, a clear contrast to the IID results can be drawn.

2

2.1 We adjusted the punctuation in some sentences to make them easier to understand, especially the sentences before and after some equation.

2.2 Your interrogation inspired us. For the same data request from the same node, TN can choose to reject the request to protect privacy. This question also concerns whether using the same dataset for different training tasks, or more specifically, for different training models, would lead to differential privacy leakage. If, for different trained models, the same data request does not lead to the leakage of differential privacy, or can be controlled, then it can be achieved to preserve the utility of a dataset that has been used in a federated learning task. If not, rejecting identical data requests is a good option to protect differential privacy. This is a good idea, and we will study this issue further in the future.

2.3

2.3.1 What we want to say is that Deep Learning is used more and more in our life and work. However, the inappropriate representation confuses readers, so we modified it.

2.3.2 We have revised this sentence according to your suggestion.

2.3.3 We modified Fig.1 as you suggested to put the individual figures in a box, and put the numbers in the box as well.

2.3.4 Such a statement really confuses readers, so we modified “public” to “define” as you suggested.

2.3.5 According your suggestion, we put the comma in the right position, and we modified the same error in any other place as well.

2.3.6 It is an Differential Privacy Mechanism. In the new manuscript, we have added symbol definitions to Table 1 on page 4 and define it. And actually it is a noisy version of f.

2.3.7 We removed the extra comma. In the new manuscript, the parameter W has been changed to G, and the element w has been changed to g but the problem still exists. We illustrate the meaning and calculation of g in the new manuscript. Actually using the parameters g and w to calculate is equivalent, but using g can reduces many unnecessary indices, This solves the problem of our index confusion to some extent. In particular, the variable G defined at last line of Page 5(l.138 in new manuscript) represents the training result received by the RN, and therefore does not contain global parameters, because the global parameters of the next round are calculated by the RN. The variable   G’ defined at l.127(before Eq.8) represents the matrix used in the RN calculation, so the two variables are different. We make the distinction clearly in the new manuscript.

2.3.8 What we described Eq.8 and Eq.9 is really far from what we want to express. In the new manuscript, we have modified the expression without changing the original scheme, and placed the coefficients in the correct positions.

2.3.9 w_{k,i}(w_{t,c} and g’_{t,c} in new manuscript) are all matrix. But for parallel computation, we reshape it into an N×1 matrix in Algorithm 3.

2.3.10 There is some ambiguity in what we do describe. In the new manuscript, we have simplified the number of indices and edited Algorithm 3 to make it clearly.

2.3.11 We delete the extra point.

2.3.12 We modify it to "Privacy budget analysis" according to your advise.

2.3.13 We put the caption below the picture, and the picture and the text no longer block each other.

Reviewer 2 Report

The paper presents a method dedicated to Kalman filter-based differential privacy federated learning. Contributions of the paper are presented in the introduction as three fields: (1) the proposal of a multiparty cooperative model training scheme that protects data privacy, (2) the proposal of a differential privacy federated learning algorithm to protect user data privacy by a differential privacy mechanism, (3) the reduction of the noise introduced by the differential privacy mechanism by Kalman filtering to improve the model’s accuracy. In my opinion, the section “Conclusion” should include larger description related to contributions mentioned in the introduction.

Author Response

Dear reviewer:

Thank you very much for your comment and professional advise. The opinions help to improve academic rigor of our article. Based on your suggestion, we have made corrected modifications on the revised manuscript. We hope our work can be improved again. Furthermore, we would like to show the details as follows:

Opinions:

1.In my opinion, the section “Conclusion” should include larger description related to contributions mentioned in the introduction.

Author response:

1.We highlight the three contributions mentioned in the Introduction in the conclusion of the article(l.224-l.228).

Reviewer 3 Report

The article submitted is to propose a method to protect privacy data. The synchronization of this paper is clear, between the introduction, methods, experiments, results and conclusions. But some things need to be improved

a. It needs to be systematic in several places, for example between lines 129 and line 130, it might be good in the arrangement if the sentence starting "There are C TN ..." until "Eq (9) is positioned before the sentence "The parameter update equation for model training is: (7). Some image positions, the placement is out of the ordinary without a clue, for example Fig.3 (p. 7) while the image is at p. 9. Although it is not far from the position, it is better to refer to Eq. 3 and its equivalent are in one sentence. However, systematics improves the readability of a piece of writing.

b. It is necessary to make consistency in writing some symbols, such as f, Sf, Cifar10 (Table 1 and Table 3), C as a set in (6), etc. Some of the symbols are not explained their function as d in (2). Then there is an equation as if it were recursive (line 111) without description.

c. Figure 2 p 8 is not referenced in the text. What is the explanation of line 153?

d. Some of the parameters listed in Algorithm 1 and Algorithm 2 need an explanation in writing, such as whether the functions are B, I, e, E.

Author Response

Dear reviewer:

Thank you very much for your comment and professional advise. The opinions help to improve academic rigor of our article. Based on your suggestion, we have made corrected modifications on the revised manuscript. We hope our work can be improved again. Furthermore, we first list the advice and then response. Besides, we also upload a pdf version.

Opinions:

1.It needs to be systematic in several places, for example

1.1 between lines 129 and line 130, it might be good in the arrangement if the sentence starting "There are C TN ..." until "Eq (9) is positioned before the sentence "The parameter update equation for model training is: (7).

1.2 Some image positions, the placement is out of the ordinary without a clue, for example Fig.3 (p. 7) while the image is at p. 9. Although it is not far from the position, it is better to refer to Eq. 3 and its equivalent are in one sentence. However, systematics improves the readability of a piece of writing.

  1. It is necessary to make consistency in writing some symbols, such as f, Sf, Cifar10 (Table 1 and Table 3), C as a set in (6), etc. Some of the symbols are not explained their function as d in (2). Then there is an equation as if it were recursive (line 111) without description.
  2. Figure 2 p 8 is not referenced in the text. What is the explanation of line 153?
  3. Some of the parameters listed in Algorithm 1 and Algorithm 2 need an explanation in writing, such as whether the functions are B, I, e, E.

Authors response:

1

1.1 Based on your suggestion, we have adjusted the position of some sentences. Adjusted descriptions are easier to understand.

1.2 We have added a description of the location when the reference location and the image location are far apart.

  1. We adjusted symbols and indexes that were inconsistently in the context (include but not limit f, Sf, Cifar10, C). We use this equation in the Algorithm 1 and it will be executed in a loop. Putting it here does confuse the reader, and it's more clearly described in line 5 of Algorithm 1. So we decided to delete the equation here.
  2. We should have used it on line 185, sorry we left it out. Now we add it on line 185.

The sample probability q is included in Eq.5, and q can be calculated from the values of the listed variables, so it is explained here. In order to reduce the confusion of readers, we have extracted the calculation method of q and put it in front of Eq.5, and deleted the content here.

  1. Based on your comments, we have added a symbol table that defines the main symbols that appear. And the minor symbols is defined in the context. This will make it clearer to the reader.
